# Iodine Deficiency, Maternal Hypothyroxinemia and Endocrine Disrupters Affecting Fetal Brain Development: A Scoping Review

**DOI:** 10.3390/nu15102249

**Published:** 2023-05-09

**Authors:** Rolf Grossklaus, Klaus-Peter Liesenkötter, Klaus Doubek, Henry Völzke, Roland Gaertner

**Affiliations:** 1Department of Food Safety, Federal Institute for Risk Assessment, D-10589 Berlin, Germany; rgrossi@web.de; 2Endokrinologikum, Center for Hormonal and Metabolic Diseases, D-10117 Berlin, Germany; kp.liesenkoetter@posteo.de; 3Professional Association of Gynecologists, D-80337 Munich, Germany; 4Study of Health in Pomerania/Clinical-Epidemiological Research, Institute for Community Medicine, University Medicine Greifswald, D-17475 Greifswald, Germany; voelzke@uni-greifswald.de; 5Medical Clinic IV, University of Munich, D-80336 Munich, Germany

**Keywords:** iodine deficiency, pregnancy, hypothyroxinemia, neurocognitive development, endocrine disruptors, brain development

## Abstract

This scoping review critically discusses the publications of the last 30 years on the impact of mild to moderate iodine deficiency and the additional impact of endocrine disrupters during pregnancy on embryonal/fetal brain development. An asymptomatic mild to moderate iodine deficiency and/or isolated maternal hypothyroxinemia might affect the development of the embryonal/fetal brain. There is sufficient evidence underlining the importance of an adequate iodine supply for all women of childbearing age in order to prevent negative mental and social consequences for their children. An additional threat to the thyroid hormone system is the ubiquitous exposure to endocrine disrupters, which might exacerbate the effects of iodine deficiency in pregnant women on the neurocognitive development of their offspring. Ensuring adequate iodine intake is therefore essential not only for healthy fetal and neonatal development in general, but it might also extenuate the effects of endocrine disruptors. Individual iodine supplementation of women of childbearing age living in areas with mild to moderate iodine deficiency is mandatory as long as worldwide universal salt iodization does not guarantee an adequate iodine supply. There is an urgent need for detailed strategies to identify and reduce exposure to endocrine disrupters according to the “precautional principle”.

## 1. Introduction

Thyroid hormones are particularly important for normal embryonal/fetal and early postnatal neurocognitive development. Depending on the severity, duration and timing of iodine deficiency in certain life stages, iodine deficiency disorders (IDDs) may be associated with physical, neurological and intellectual deficits in humans. Severe iodine deficiency during pregnancy can lead to a number of adverse effects on maternal and child health, including goiter, hypothyroidism, stillbirth, increased neonatal mortality, neurological damage, and mental impairment [1]. In addition, global exposure to endocrine-disrupting chemicals (EDC) is increasing [2,3,4]. Exposure to these chemicals in the presence of inadequate iodine supply might be additionally harmful to the embryonal/fetal and neonatal brain development, growth, differentiation, as well as metabolic processes in adulthood [5,6,7,8,9,10,11,12]. 

Both iodine deficiency and exposure to EDCs have a negative impact on general health and on socio-economic systems. Annual costs for seven categories of EDCs with the highest causality have been estimated to be at least €33.1 billion in Europe. The largest proportion of costs is related to the loss of IQ points and neurocognitive diseases [13,14,15,16,17]. In addition, there is growing evidence that exposure to EDCs, including air pollution, affects not only the development of brain function [10,18,19,20,21] but also the outcomes of pregnancy and childbirth [22,23,24,25,26].

The “endemic goiter” has long been a synonym for iodine deficiency, and the aim has always been to prevent the enlargement of the thyroid gland and overt thyroid dysfunction. Within the last decades, however, the consequences of mild to moderate iodine deficiency on the cognitive development of the embryo/fetus have also come into focus [27]. 

Epidemiological and experimental studies of mild to moderate iodine deficiency over the past two decades have shown that embryonal/fetal brain development can be impaired not only in hypothyroid mothers but also in hypothyroxinemic mothers in the early stages of pregnancy [28,29,30,31,32]. Subtle changes in fetal brain development were observed even with maternal thyroid hormone levels within the lower reference range, although the results are not homogeneous, and, therefore, isolated maternal hypothyroxinemia (IMH) is not yet generally accepted as an independent thyroid disease. Instead, it is assumed to be the result of iodine deficiency and/or EDC contamination or of other factors, as extensively summarized and discussed by others [6,33,34,35].

The aim of this scoping review is to identify gaps in knowledge about the effects of mild and moderate iodine deficiency and additional environmental influences such as EDCs and air pollution during pregnancy on the cognitive and psychosocial development of the offspring and how significant IMH might be. Above all, the results of systematic reviews should be used to present the need for a safe and secure iodine supply to political decision-makers.

## 2. Materials and Methods

This study is a scoping review in accordance with the PRISMA Extension for Scoping Reviews (PRISMA-ScR) statement [36]. The study selection process is documented in Figure 1. Literature searching was performed in PubMed, Medline, Cochrane, Web of Science, Google Scholar, World Health Organization (www.who.int/en/, accessed on 1 April 2022), and Iodine Global Network (https://www.ign.org, accessed on 1 April 2022) databases between January 1990 and April 2022. Keywords were iodine, pregnancy, thyroid hormone, thyroid diseases, endocrine disruptors, hypothyroxinemia and subclinical hypothyroidism, searched for in combination using AND and OR operators. Inclusion criteria were observational and interventional studies, including reviews and meta-analyses, which assessed the iodine status of pregnant women in Europe; assessed thyroid hormone concentrations (fT4, TSH) during the first trimester of pregnancy; iodine supplementation before and during pregnancy and perinatal exposure of certain EDCs, which were limited to potential thyroid-disrupting chemicals (TDCs) such as organochlorine pesticides (OCPs), polychlorinated biphenyl compounds (PCBs, OH-PCBs), perchlorate, thiocyanate, nitrate, phthalates, genistein, 4-nonylphenol (NP), benzophenone 2 (BP2), amitrole, polybrominated diphenylethers (PBDEs), triclosan and bisphenols. 

The initial search retrieved a total of 4920 articles. All duplicate papers were double-checked and excluded. Titles and abstracts were screened for relevance independently by two reviewers (RGr and KPL), with any disagreements being resolved by discussion and involvement of a third reviewer (RGa) where necessary. The full text of potentially relevant papers was retrieved and screened in the same way using the prespecified inclusion and exclusion criteria. Additional records were identified from the reference lists of retrieved publications or through other sources. A total of 279 articles remained eligible for inclusion in this study. The results were discussed with all authors during virtual meetings.

## 3. Results and Discussion

### 3.1. Iodine Requirements and Iodine Status of Pregnant Women in Europe

Pregnant women have an approximately 50% higher iodine requirement compared to non-pregnant women due to increased thyroid hormone production, increased renal iodine clearance and transplacental transmission of iodine to the fetus [37]. Accordingly, the recommendation for mean iodine intake during pregnancy is 150–249 µg/day [38].

The WHO recommends four possible markers to evaluate the iodine intake of a defined population: urinary iodine concentration (UIC), the serum concentration of thyroid stimulating hormone (TSH), serum concentration of thyroglobulin (Tg) and the thyroid volume [39], but not free thyroxine (fT4). Mean UIC is used in most epidemiological studies because of the easy practicability that testing involves. A mean UIC of 250 µg/L during pregnancy is assumed as an adequate iodine supply [39,40]. However, no epidemiological studies so far have compared the four parameters to find the best method.

It is estimated that worldwide, approximately 54% of women have insufficient iodine supply during pregnancy [41]. Within Europe, this percentage is even higher; more than 70% (*n* = 21) of the 29 European countries have an insufficient iodine intake (Table 1) [42,43,44,45,46,47,48,49,50,51,52,53,54,55,56].

The reason is that most European women of childbearing age live in countries with voluntary household salt iodization. As a result, the median UIC in these countries is below 100 μg/L [57]. Recently, national representative data from children and adolescents in Germany revealed a mean UIC of 89 µg/L [55,58]. Iodine intake has clearly decreased in the last decade compared to the baseline survey (2003–2006). The cause might be that the use of iodized salt in artisanal and industrially produced foods has declined, and vegetarian and vegan diets have been attracting more people, particularly young people [59]. 

A median UIC > 150 μg/L was reported in only a few EU states with mandatory universal salt iodization programs, such as Bulgaria or Romania (see Table 1). In Poland and Italy, iodization of table salt is mandatory but not allowed in processed foods, with the exception of formulations for toddlers [60,61,62]. Belgium, Denmark and the Netherlands have introduced the mandatory use of iodized salt in bread. However, this strategy does not seem to meet women’s higher iodine requirements during pregnancy [63,64]. Additional iodine supplementation for women of childbearing age might be necessary in most European countries [40,65,66]. The trend of a re-emergence of iodine deficiency among vulnerable groups, such as reproductive-age women in most European countries, appears to mirror the trends in other industrialized nations [67].

### 3.2. Isolated Maternal Hypothyroxinemia (IMH)

#### Definition, Prevalence and Causes

The first study described IMH during pregnancy as the free maternal serum thyroxine concentration (fT4) being below the 10th percentile with a TSH value in the reference range [68]. However, there are significant differences in the definition of IMH (fT4 below 10th or 2.5th percentile) in later studies that make the interpretation of the data difficult [69,70,71]. The prevalence of IMH varies between 1% and 25%, depending mainly on the diagnostic criteria, the trimester of pregnancy or the method of fT4 measurement. In countries with iodine deficiency, the frequency of IMH is expected to be higher [33,72]. IMH might be due primarily to mild or moderate iodine intake, which is associated with an increased release of triiodothyronine (T3) and decreased T4 to save iodine. As only fT4 is actively transported through the placenta in the first 4 months, IMH might affect embryonic brain development before the fetal thyroid function starts [73]. 

Prenatal exposure of maternal and embryonal/fetal thyroid to EDCs might additionally influence the brain development of the offspring [6,7,74]. Ethnicity, mother’s age, parity, pre-pregnancy body mass index and vitamin D deficiency (as potential causative factors) are also associated with IMH [33,75,76,77,78,79,80]. It should also be considered that the shortage of trace elements beyond iodine, e.g., selenium and iron, are particularly important for thyroid function and should also be viewed as risk factors of IMH [81,82,83].

ATA guidelines do not recommend fT4 analysis in pregnancy but only total T4 [69] measurement. The recommended method to assess fT4 in pregnancy would be dialysis or ultrafiltrate of serum samples employing liquid chromatography-tandem mass spectrometry (LC/MS/MS) [69,70,84]. This, however, has never been done in epidemiologic studies. 

The most recent clinical guidelines, therefore, regard population-based, trimester-specific reference ranges for serum TSH and T4 levels in a local, euthyroid, pregnant population as the gold standard [69,85]. 

### 3.3. Prenatal Brain Development, Timing of Thyroid Hormone Action and Identification of Specific Thyroid-Related Modes-of-Actions (MoAs) in Connection with EDCs

The embryonal and early fetal brain development depends on the maternal T4 (Figure 2) [74,86,87,88,89,90]. Only fT4 enters the embryonal/fetal blood–brain barrier. Exposure to some ECDs has been shown to interfere with the thyroid system at multiple sites affecting the developing brain, for example, by inhibiting deiodinase activities as well as with transport mechanisms like transthyretin (TTR) in the placenta, in the embryo/fetus [6,9,11,74]. 

Identifying this early, critical period can have direct clinical implications for risk assessment and the window of opportunity for treatment (s. Figure 2). During this critical phase of development, reduced maternal placental fT4 transfer most likely has the greatest impact on a child’s neurological development [91,92,93,94,95]. Brain MRI imaging enables objective measurements of brain development and provides detailed information about brain structures. Imaging data provide information on neurogenic processes during certain stages of fetal brain development [96,97].

The results of part of the longitudinal Generation R study in Rotterdam, including 1981 mother-child pairs, indicate a “fetal programming effect”. Measurements of maternal TSH and fT4 in early or middle pregnancy (≤18 weeks) were compared with MRI data from the brains of children aged 10 years. It was found that there was an inverted U-shaped association of maternal TSH with the total volume of gray matter in the offspring and with the volume of cortical gray matter. It was also shown for the first time that this association with a later occurrence of neurodevelopmental disorders is more obvious when thyroid function is measured before the 14th week of pregnancy. Both low and high maternal thyroid function, particularly in early pregnancy, were shown to be associated with smaller child total grey matter and cortical volume [98,99]. However, no causal relationship between maternal thyroid function and morphological brain changes in the child has yet been conclusively clarified. Further research is warranted to determine how MRI, as a morphometric method, can help to assess the effects of maternal thyroid function on the child’s cognitive function [100,101].

There is increasing evidence that TDCs also interfere with several intracellular thyroid hormone actions and brain development [6,9,11,102,103,104,105,106,107,108]. Obviously, iodine deficiency might promote these deleterious effects and thus deregulate transcription and thyroid hormone-induced epigenetic effects on target genes [109]. The urgency of this issue is the coincidence of the still prevailing inadequate iodine supply and continuously increasing exposure of humans to TDCs [6,22,110,111,112]. 

TDCs affect pregnancy not only by acting as hormonal agonists/antagonists but also indirectly by disrupting maternal, placental, and fetal homeostasis. This might be mediated by an impact on oxidative stress, the hormonal milieu, metabolomic profile and microbiome [22,113,114,115,116,117,118,119,120]. It is believed that the adverse health effects in offspring caused by EDCs, including air pollution, can be caused by two mechanisms: first, directly through the placenta and thus to the fetal circulation and/or second, indirectly through oxidative stress of the placenta, inducing inflammation and epigenetic changes in the placenta as well as in the offspring [10,121,122,123,124,125,126]. 

Epidemiological studies on physiological fluids collected in pregnant women (blood, serum, urine, amniotic fluid, placenta) show that combined exposure to EDCs (“cocktails”) is commonplace and widespread [127,128,129]. It is unlikely that safe levels of EDC contamination can be defined because of the diverse actions of EDCs like low dose effects, possible non-linear dose responses, cumulative effects often expected from combined exposure and trans-generational effects with different effects during vulnerable exposure windows [16,130,131,132,133,134].

Some of the TDCs’ effects on TH metabolism are well characterized and summarized in Table 2 [135,136,137,138,139,140,141,142,143,144,145,146,147,148,149,150,151,152,153,154,155,156,157,158,159,160,161,162,163,164,165,166].

Ortho-substituted polychlorinated biphenyl (PCP) congeners (95 or 101) decrease pituitary response to thyrotropin-releasing hormones [167]. Perchlorate, thiocyanate and nitrate competitively inhibit the iodide uptake by the NIS. This might also be true for benzophenone 2 [102].

Animal studies have indicated that phthalates alter thyroid signaling through a number of potential mechanisms, including interference with the TSH receptor, binding to transport proteins, interfering with the hypothalamic-pituitary-thyroid axis, and changing NIS-mediated iodide uptake, iodothyronine deiodinases, or hepatic enzymes [102,168].

PCPs and their metabolites, as well as polybrominated diphenyl ethers (PBDEs), also bind TTR and displace T4 [7,102,104,105]. TTR has been proposed as being of importance by transferring thyroid hormones across the blood–brain barrier as well as via placenta to the fetal compartment. Another mechanism for the decrease in serumT4 concentrations in both adults and neonates may be the ability of PCBs to induce hepatic microsomal enzymes leading to biliary excretion and elimination of the thyroid hormones. Such changes could be particularly adverse during early pregnancy and even have long-term effects on the brain, as demonstrated in Table 2 [3,153,154,155,156,157,169,170].

Bisphenols, including BPA, can interfere with thyroid hormone synthesis, transport and metabolism. The main mechanism of action is thought to be the binding of BPA to TR and interference with thyroid hormone [166]. Human data during pregnancy substantiate experimental findings suggesting that BPA could potentially affect thyroid function and deiodinase activities in early gestation (6–14 weeks). BPA was associated with a lower ratio of both fT4/fT3 and TT4/TT3 (total thyroxine/total triiodothyronine), as well as a lower TT4 concentration [171]. BPA at environmentally-relevant doses also has epigenetic actions that can lead to heritable changes in gene expression. BPA might alter DNA methylation quite early in embryonic development and may impact the chromatin state of germ cells [172].

Whatever the level at which the disruption occurs, it can result in decreased T3 binding to nuclear TRs. This might then modulate transcriptional activity and induce epigenetic disruption [9,173,174,175,176,177]. Changes in TH availability—and thus in DNA methylation rates—during organogenesis and developmental transitions can not only increase the risk of a low IQ but also lead to other long-term offspring outcomes such as cardiometabolic, neurodevelopment and behavioral defects [9,92,109,175,178].

### 3.4. Mild and Moderate Iodine Deficiency and Its Consequences

In contrast to severe iodine deficiency, a mild to moderate intrauterine iodine deficiency has more subtle but nonetheless important long-term cognitive and psychosocial consequences for the offspring [179]. 

In observational studies about adverse effects on cognitive development and behavioral disorders related to mild iodine deficiency, maternal blood samples were usually taken between the ninth and 13th week of pregnancy (Table 3). The neurological examinations of the offspring were carried out between the ages of 6 months and 16 years [95]. Overall, the study designs were very different. The differences relate to the criteria for the selection of mother-child pairs, to the reference values and ranges used to determine the different degrees of maternal hypothyroidism and/or hypothyroxinemia, and to the different tests for neurological development used (s. Table 3).

All studies, except one by Grau et al. [187], who examined the effects of low maternal fT4 levels towards the end of the first trimester, reported impairment of cognitive and motor development in exposed children [30,34,69,91,93,181,182,183,188,189,190]. With the progress of pregnancy, the correlation weakened gradually and disappeared until late pregnancy [32,186,191]. 

Two systematic reviews [192,193] and five meta-analyses [180,194,195,196,197] evaluated the association between maternal iodine deficiency and intellectual outcomes in the offspring. In general, these publications suggested an association between high TSH and/or low T4 levels in the serum of pregnant mothers and impaired neurological development and behavioral problems in the child.

Using the random effects model, two meta-analyses [180,195] revealed that subclinical hypothyroidism and hypothyroxinemia in mothers are associated with indicators of intellectual disability in the offspring (odds ratio [OR] 2.14, 95% confidence interval [CI] 1.20 to 3.83, *p* = 0.01 and OR 1.63, 95% CI) 1.03 to 2.56, *p* = 0.04) [180]. On the other hand, the results for behavioral disorders like ADHD or autism are inconsistent and require further studies [180,197]. The meta-analysis by Levie et al. [197], who evaluated data from three cohort studies together, found no clear indications for a connection between maternal TSH and fT4 up to the 18th week of pregnancy and ADHD in the offspring [197]. The systematic review by Drover et al. [193], which included 28 studies (5 of which looked at thyroid hormone levels in newborns), reported moderate evidence of such an association between IMH and ADHD.

Overall, none of the systematic reviews and meta-analyses showed clear cut-offs for high TSH and/or low fT4 levels in the serum of pregnant mothers, which indicates an increased risk of neurodevelopmental disorders in children. Such cut-offs could not be determined because the epidemiological studies were not designed to show quantitative limits (s. Table 3).

### 3.5. Influence of TDCs, Including Air Pollution on Embryonic/Fetal Neurodevelopment in Iodine-Deficient Areas

So far, studies investigating maternal hypothyroxinemia due to mild to moderate iodine deficiency have not considered additional prenatal exposure to TDCs (s. Table 2, right column). However, retrospective case-control and cohort and population studies linking TDC exposure with epidemiological data on thyroid hormone-related (dys-)functions provide clear evidence that the development of the embryonal/fetal and neonatal brain as well as growth, differentiation and metabolic processes are at risk of suffering adverse TDCs effects [6,198]. In recent years, there has been a significant increase in neurodevelopmental disorders, including autism and ADHD [2,12,199,200,201,202].

Public health concern exists for mildly iodide-deficient pregnant women who are exposed to perchlorate, thiocyanate, nitrate or other environmental antithyroid agents [4,8,11,16,203,204,205,206,207,208,209]. In a dose-response model between iodide and perchlorate exposure in food, it was shown that a low iodide intake of 75 μg/day and a perchlorate daily dose of 4.2 μg/kg are sufficient to induce hypothyroxinemia, while an adequate iodine intake of 250 μg/day a higher perchlorate daily dose of around 34 μg/kg is required [210]. Iodine supplementation would be sufficient to prevent the goitrogenic effects of perchlorate exposure at current regulatory limits among at-risk individuals [205]. Iodine deficiency could therefore deteriorate the effects of TDCs exposure, especially during pregnancy [4,8,11,16,104,105].

Certain phthalates, including di-(2-ethylhexyl) phthalate (DEHP) and di-n-butylphthalate (DnBP), have antithyroid activity occurring through several possible mechanisms, such as down-regulation of NIS and interacting with hormone synthesis-related proteins, deiodinases, TTR, receptors, and hepatic enzymes [6,9,211,212]. Because phthalates may have multiple and possibly overlapping targets in the HPT axis, sometimes acting as an agonist or antagonist, the outcome from a given phthalate blend may not be predictable. For example, it had been shown in the Norwegian mother, child and father cohort study (MoBa) that exposure to certain phthalates in pregnant women increased TT3 and FT3, but only when iodine intake was low (<150 µg/d), whereas in those women with high iodine intake (>150 µg/d), TSH increased, and TT4 and FT4 decreased [213].

Furthermore, epidemiologic evidence suggests that prenatal exposure to phthalates is associated with emotional and behavioral difficulties in children [9,145,146,147,148,149,214,215]. However, a recent literature review including 17 epidemiological studies reveals no clear pattern of association between maternal exposures to phthalates during pregnancy and offspring neurodevelopment. This, again, might be caused by inconsistent study protocols, test systems and confounders [216,217].

A prospective pregnancy and birth cohort study examined BPA interaction with thyroid hormones in pregnant women and newborns. Higher BPA exposure is associated with decreased TSH in umbilical cord serum in girls. BPAs have the greatest negative effects on girls born to mothers with iodine deficiency [218]. A birth cohort study in China showed that the concentration of BPA in urine in the prenatal period was associated with low TSH in overweight mothers, but there was no association with fT4, fT3 and TSH in umbilical cord serum [219]. The disturbance of thyroid hormone (TH) levels as a result of prenatal exposure to BPA may be associated with long-term neurobehavioral changes at a later age [220,221,222]. It should be noted that there are various kinds of test batteries for child neurodevelopmental assessment at different ages whose findings have been inconsistent among studies. In addition, the timing and number of exposure assessments have varied. However, ADHD symptoms, especially among boys, constantly suggested an association with both prenatal and concurrent exposure to BPA [223]. Although there is limited evidence on the adverse effects of prenatal and postnatal BPA exposures, pregnant women and young children should be protected from exposure based on a precautionary approach [224,225].

Furthermore, a pilot study suggests that 2,3′,4,4′,5-pentachlorobiphenyl (PCB 118) has a negative impact on neurocognitive development and probably reduces the benefits of iodine supplementation in areas with borderline iodine deficiency. Therefore, TDC exposure should be considered when designing studies on the benefits of iodine supplementation during pregnancy [226].

From observational studies, there is strong evidence that TDCs are disturbing the thyroid hormone metabolism (s. Table 2). This is reported for several classes of TDCs, for mother-child pairs, in case controls, smaller cohorts or larger epidemiological studies [6,102,135,136,137,138,139,140,141,142,143,144,145,146,147,148,149,150,151,152,153,154,155,156,157,158,159,160,161,162,163,164,165,166,167,168,169,170,171,172,173,174,175,176,177,178,179,180,181,182,183,184,185,186,187,188,189,190,191,192,193,194,195,196,197,198,199,200,201,202,203,204,205,206,207,208,209,210,211,212,213,214,215,216,217,218,219,220,221,222,223,224,225,226,227,228,229,230,231,232]. However, further research and long-term clinical studies are necessary to finally elaborate on the dose-related associations [6,168,227,229,233,234,235,236].

Air pollution is a leading risk factor for the global disease burden, but the negative effects of exposure to particulate matter <2.5 μm (PM_2.5_) during pregnancy have not been considered in the past [237,238,239]. However, there is growing evidence of the negative effects of exposure to burn-related air pollution on the neurological development of fetuses and childhood behavior [20,240,241,242]. Air pollution may interfere with maternal thyroid function during early pregnancy, as shown in cohort studies from four European cohorts [10] and in Shanghai [243]. A 10 mcg/m^3^ increase in PM_2.5_ exposure in both the first and second trimester was associated with 28% (OR = 1.28, 95% CI, 1.05–1.57) and 23% (OR = 1.23, 95% CI, 1.00–1.51) increases in the odds of maternal hypothyroxinemia, respectively [243]. However, both studies have some limitations. Neither the iodine concentration in the urine of the pregnant women [244] nor the exposure to other environmental chemicals was considered [245].

The available evidence suggests that intrauterine PM_2.5_ exposure can alter prenatal brain development through oxidative stress and systemic inflammation, leading to chronic neuroinflammation, microglial activation, and neuronal micturition disorder [18,168,246]. It has been shown that particulate matter exposure during fetal lifetime was associated with structural changes in the child’s cerebral cortex, as well as with impairment of essential executive functions, such as inhibitory control [247,248].

Studies that focused on exposures to air pollution, especially PM and NO_2_, during the prenatal period and the first years of life found associations with reduced psychomotor development [249,250] and impairment in cognitive development [19,251,252], as well as with autism-spectrum disorders [201,253,254,255]. However, these results could not be confirmed by others [20,21,256,257,258].

### 3.6. Prevention and Treatment of IMH

Since studies on the effects of IMH on cognitive and motor development, as well as on the risk of neuropsychiatric diseases in children, show a clear connection to early pregnancy; the central clinical question remains whether these complications can be prevented by early iodine supplementation or L-Thyroxine substitution [29,33].

Treatment of IMH or subclinical hypothyroidism with L-Thyroxine during early pregnancy revealed no benefit concerning the neurodevelopment of the children at the age of 6 and 9 years. However, L-Thyroxine supplementation started at the mean of the 12th week of pregnancy, which is too late [259,260]. This is why the ATA guidelines do not recommend L-Thyroxine supplementation [69]. However, based on new epidemiological data, ETA guidelines are considering L-Thyroxine supplementation during the first trimester rather than later [85]. The results of a recent study showed that early L-Thyroxine supplementation in women with TSH levels of >2.5 mU/L and fT4 < 7.5 pg/mL at or before the ninth gestational week (GW9) is safe and improves the progress of gestation. Whether the neurodevelopment of these offspring also improved, however, has not been studied so far. These data support the recommendation to adopt these cut-off levels for L-Thyroxine supplementation, which should be started as early as possible, ideally before the end of the first trimester of gestation, and TSH suppression should be avoided [261].

In regions with mild to moderate iodine deficiency, iodized salt intake, regularly used at least 24 months before pregnancy, can significantly improve maternal thyroid economy and reduce the risk of maternal thyroid insufficiency during pregnancy. This is probably due to a restoration of intrathyroidal iodine stores [262,263,264,265,266,267,268]. The importance of this finding is supported by the results of a large prospective cohort, including mothers and offspring. A positive association between preconception maternal iodine status and the cognitive function of the offspring at the age of 6–7 years could be demonstrated [265]. In contrast, meta-analyses of iodine supplementation starting during pregnancy found no effect on child neurodevelopment [196,264,269,270,271,272]. The lack of beneficial effects of iodine supplementation, typically after the first trimester, bypasses the critical period of development early in gestation.

There is some concern that over-the-counter iodine-containing supplements might contain high doses that temporarily disturb thyroid hormone production and/or release. Therefore, moderate iodine deficiency should be prevented already before conception [273,274]. Well-designed randomized controlled trials investigating a daily supplementation with 150–200 µg iodine in preconception, pregnancy, and lactation are underway to investigate children’s neuropsychological development [275,276,277,278]. Also, more data are needed to determine optimal and safe upper limits of iodine supplementation in pregnant women and assess the potential risks of chronic high iodine intake during pregnancy [270,279].

The Krakow Declaration of Iodine, published by the EU thyroid consortium and other organizations, raised major points about how iodine deficiency can be efficiently eradicated in Europe. It was demanded that (1) universal salt iodization should be harmonized across European countries, (2) regular monitoring and evaluation studies have to be established to continuously measure the benefits and potential harms of iodine fortification programs and (3) societal engagement is needed to warrant sustainability of IDD prevention programs [280].

## 4. Conclusions

A paradigm shift has occurred in assessing the epidemiology of IDD away from goiter and focusing on the iodine status of pregnant women. This is still insufficient in most European countries, including Germany. Mild to moderate iodine deficiency, as well as probable IMH in early pregnancy, might have long-term negative effects on the cognitive and psychosocial development of children. Population-based, trimester-specific reference ranges for serum TSH and fT4 levels need to be used for the diagnosis.

Effective iodine prophylaxis for negative effects on the cognitive and psychosocial development of children should be carried out preconceptually, as the adverse effects on the developing brain are caused by reduced availability of maternal fT4 in the first weeks of pregnancy. Effective prevention through iodine supplementation for risk groups, including women of childbearing age, should, therefore, take place before conception as long as there is no nationwide universal salt iodization.

Adequate iodine supply needs to be achieved globally as this may reduce some adverse effects of additional exposure to TDCs. According to the “precautionary principle” of reducing the risk even in the absence of causal evidence, measures should be taken to reduce the environmental impact of TDCs.

## Figures and Tables

**Figure 1 nutrients-15-02249-f001:**
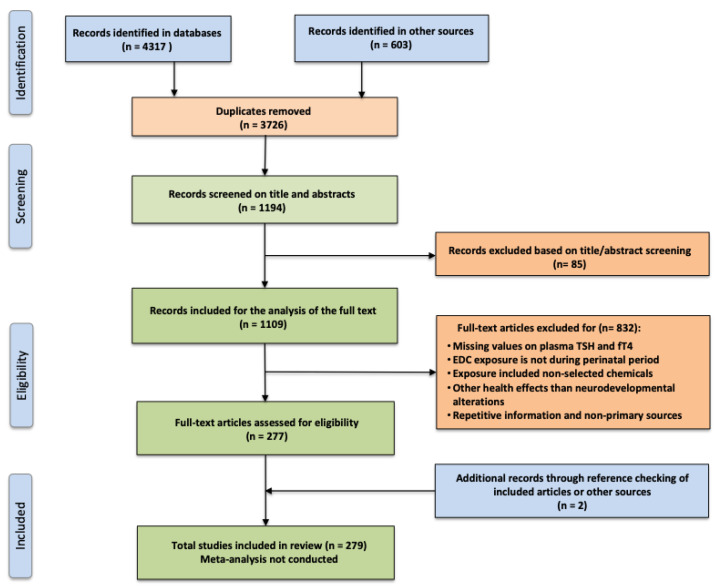
The PRISMA diagram showing the search strategy and inclusion/exclusion criteria at each step.

**Figure 2 nutrients-15-02249-f002:**
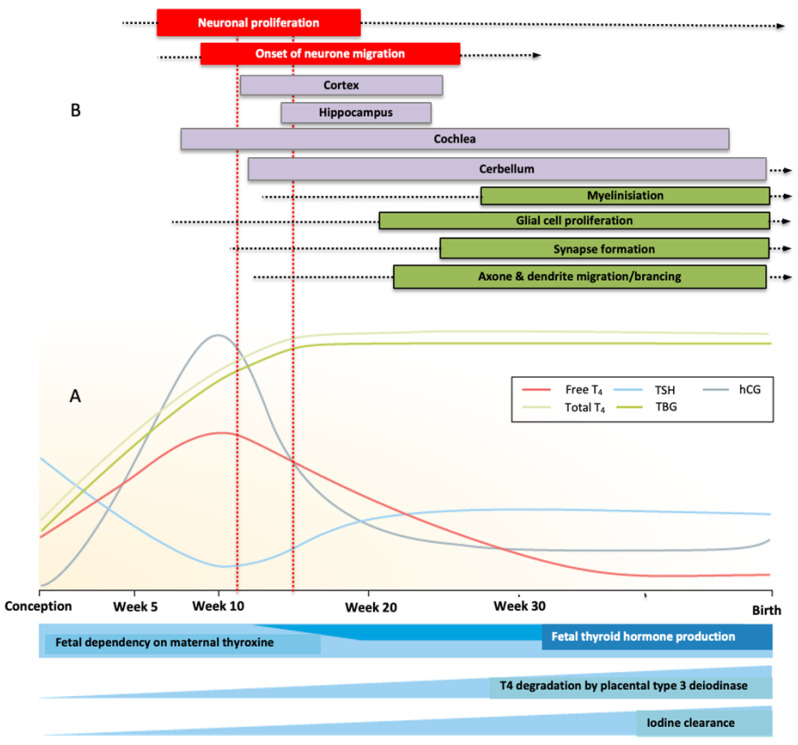
Changes in thyroid physiology during pregnancy (**A**) and the relationship between thyroid hormone activity and brain development (**B**) (adapted from [88,89]. (**A**) shows the main changes in thyroid physiology during pregnancy. An increase in thyroxine-binding globulin (TBG), an increase in thyroxine consumption by the fetus, and an increase in placenta type 3 deiodinase expression require upregulation of thyroid hormone production in order to maintain sufficient availability of thyroid hormones. These upregulations are largely mediated by increased thyroid stimulation by human chorionic gonadotropin (hCG), which ultimately leads to a net increase in free T4 concentration and a subsequent decrease in TSH concentration. (**B**) shows the relationship between thyroid hormone activity and brain development. Note the time window (between the two red dotted lines) in which a drop in maternal thyroid hormone (fT4) has a particular impact on neural proliferation and migration, and development of the inner ear. For further explanations, see the text.

**Table 1 nutrients-15-02249-t001:** Iodine intake in the general population and in pregnant women in Europe.

Country	General Population ^a^				Pregnant Women ^b^			
	Median (UIC) (μg/L)	Date of Survey (N, S)	Population	Population Iodine Intake	Median (UIC) (μg/L)	Date of Survey (N, S)	Iodine Intake	Legislation Status ^e^ (Year)
Austria	111	2012 (N)	SAC (7–14)	Adequate	87	2009–2011 (S)	Insufficient	Mandatory (1999)
Belgium	113	2010/11 (N)	SAC (6–12)	Adequate	124	2010 (N)	Insufficient	Voluntary (2009)
Bulgaria	182	2008 (N)	SAC (7–11)	Adequate	165	2003 (N)	Adequate	Mandatory (2001)
Croatia	248	2009 (N)	SAC (7–11)	Adequate	140	2009, 2015 (S)	Insufficient	Mandatory (1996)
Denmark	145	2015 (S)	SAC	Adequate	101	2012 (S)	Insufficient	Mandatory (2000) ^f^
Finland	96	2017 (N)	Adults (25–74)	Insufficient	115	2013–2017 (S) ^f^	Insufficient	Voluntary ^f^
France	136	2006–2007 (N)	Adults (18–74)	Adequate	65	2006–2009 (S)	Insufficient	Voluntary
Germany	89	2014–2017 (N)	SAC, Adolescent (6–12)	Insufficient	54	2008–2011 (N) ^c^	Insufficient	Voluntary
Greece	132	2018 (N)	Adults	Adequate	127	2008–2015 (S)	Insufficient	Voluntary
Hungary	228	2005 (S)	SAC (10–14)	Adequate	128	2018 (S) ^d^	Insufficient	Mandatory (2013)
Ireland	111	2014–2015 (N)	Adolescent girls (14–15)	Adequate	107	2008–2010 (S)	Insufficient	Voluntary
Italy	118	2015–2019 (S)	SAC	Adequate	72	2002–2013 (S)	Insufficient	Mandatory (2005)
Netherlands	130	2006 (S)	Adults (50–72)	Adequate	223	2002–2006 (S)	Adequate	Voluntary
Poland	112	2009–2011 (S)	SAC (6–12)	Adequate	113	2007–2008 (S)	Insufficient	Mandatory (2010)
Portugal	106	2010 (N)	SAC	Adequate	85	2005–2007 (N)	Insufficient	Voluntary
Romania	255	2015–2016 (N)	SAC (6–11)	Adequate	206	2016 (S)	Adequate	Mandatory (2009)
Spain	173	2011–2012 (N)	SAC	Adequate	120	2002–2011 (S)	Insufficient	Voluntary
Sweden	125	2006–2007 (N)	SAC (6–12)	Adequate	98	2006–2007; 2010–2012 (S)	Insufficient	Voluntary (1936) ^f^
Switzerland	137	2015 (N)	SAC (6–12)	Adequate	136	2015 (N)	Insufficient	Voluntary
United Kingdom	166	2015–2016 (N)	SAC, Adolescent (4–18)	Adequate	99	2002–2011 (S)	Insufficient	no USI-program

Abbreviations: SAC, School-age children (typically 6–12 years old); UIC, urinary iodine concentration; USI, Universal salt iodization; N—Nationally-representative data; S—Sub-national data only; Dates according to ^a^ [56], ^b^ [42], ^c^ [57], ^d^ [43], ^e^ [49], ^f^ [50,51].

**Table 2 nutrients-15-02249-t002:** Potential thyroid-disrupting chemicals (TDCs) targeting thyroid signaling pathway.

Examples of Chemicals	Target of TDCs Action and Results	Neurodevelopmental Alterations
Organochlorine pesticides (OCPs) ^1^ Polychlorinated biphenyl compounds (PCBs) ^2^	**TSH receptor signaling** and decreased stimulation of thyrocytes [105].	Impaired cognitive, motor and communication development [135,136,137,138,139] Impaired cognitive, motor development and playing activity [140] Reduced IQ [141] Development of ADHD-associated behavior [142]
Perchlorate ^3^ Thiocyanate ^3^ Nitrate ^3^ Phthalates ^4^	**Na+/I− symporter (NIS)** and inhibition of TH biosynthesis.	Impaired cognitive development [143] Pre- and postnatal exposures to tobacco may influence neurocognitive development [144] Sex-specific effects on cognitive, psychomotor, and behavioral development [145,146,147] Lower nonverbal and verbal IQ scores in offspring [148,149]
Propylthiouracil (PTU) Methimazole (MMI) Genistein ^5^ 4-Nonylphenol (NP) ^6^ Benzophenone 2 (BP2) ^7^ Pesticide (Amitrole) ^8^	**Inhibition of thyroid peroxidase (TPO)** results in decreased TH synthesis and subsequent reduction in circulating concentrations of THs.	Increased risk of periventricular heterotopia [150] TH Insufficiency Induces brain malformation and learning impairments [151] Decreased cognitive function [152]
OH-PCBs ^2^ Polybrominated diphenyl ethers (PBDEs) ^9^ Phthalates ^4^ Genistein ^5^	**TH-distributor proteins**: displacement of T4 and T3 by thyroid serum binding protein transthyretin (TTR) and/or thyroid binding globulin (TBG) disturbs the TH homeostasis and decreases plasma TH levels.	Impaired cognitive, behavioral, and motor development [153,154,155,156,157] Delayed neurodevelopment [158]
Polychlorinated biphenyls (PCBs, OH-PCBs) ^2^ Triclosan ^10^	Upregulation of **thyroid hormone catabolism** via the activation of hepatic nuclear receptors leads to a decrease in circulating TH levels [93,148].	Impaired early motor development [159] Hearing loss [160] Altered serum thyroid hormone levels [161,162]
Silymarin ^11^	Interference with cellular **transmembrane transporters** (MCT8, MCT10 and OATP1C1) inhibits T3 uptake.	Unwanted effects on the TH axis [163]
Erythrosine ^12^ 6 propylthiouracil PCBs ^2^	Modification of the **deiodinase enzyme activities** (DIO2, DIO3) by competitive inhibition of the enzyme or by interaction with its sulfhydryl cofactor.	With the exception of FD&C Red No. 3 dye, which causes thyroid tumors in rats, no studies to date have shown that chemicals that affect DIO expression and/or activity directly manifest in undesirable outcomes [105].

^1^ OCPs—especially used in agriculture to protect cultivated plants, but due to both their environmental persistence and neurotoxicity, their use has been banned or greatly reduced in the last decades. ^2^ PCBs—banned compounds used to make electrical equipment like transformers and in hydraulic fluids, heat transfer fluids, lubricants and plasticizers. ^3^ Perchlorate, thiocyanate, and nitrate—Individuals are exposed to these contaminants through food or other sources (e.g., cigarette smoke for thiocyanate or rocket propellant and fertilizers for perchlorate and nitrate). ^4^ Phthalates—used to make plastics more flexible, they are also found in some food packaging, cosmetics, children’s toys, and medical devices. ^5^ Genistein—a naturally occurring substance in plants with hormone-like activity found in soy products like tofu or soy milk. ^6^ 4NP—used in manufacturing antioxidants, lubricating oil additives, laundry and dish detergents, emulsifiers, and solubilizers. ^7^ BP2—is no longer permitted as a UV filter to be used in sun lotions within, for example, the European Union. However, it is still contained in plastic materials or many cosmetics to prevent UV-induced damage. ^8^ Amitrole—used as herbicide. ^9^ PBDEs—used to make flame retardants for household products such as furniture foam and carpets. Though most PBDEs have been banned or are being phased out, they are still persistent in the environment. ^10^ Triclosan—may be found in some anti-microbial and personal care products, like liquid body wash. ^11^ Silymarin—flavonoid mixture, a purified extract of the milk thistle. ^12^ Erythrosine, also known as Red No. 3, is an organoiodine compound. It is a pink dye that is primarily used for food coloring.

**Table 3 nutrients-15-02249-t003:** Observational studies about adverse effects on cognitive development and behavioral disorders related to mild iodine deficiency—characteristics of all studies included in the systematic review [180]. (Sibling papers merged).

Author, Year [Reference]	Total Number of Participants Tested for Outcome	Country	Maternal Thyroid Dysfunction	Gestation at TFT	Criteria for Thyroid Dysfunction	Child Age at Assessment	Neurodevelopmental Outcome Measures
Pop et al., 1999 [68]	220	The Netherlands	HR	12 and 32 weeks	10th percentile fT4 (<10.4 pmol/L) and fifth percentile fT4 (<9.8 pmol/L)	10 months	Bayley Scales of Infant Development
Pop et al., 2003 [181]	125	The Netherlands	HR	12, 24 and 32 weeks	fT4 <10th Percentile (12.10 pmol/L)	1–2 years	Bayley Scale of Infant Development
Kasatkina et al., 2006 [95]	35	Russia	HR	1st and 3rd trimesters	fT4 < 12.0 pmol/L	6, 9 and 12 months	Gnome method, in particular, the Coefficient of Mental Development
Li et al., 2010 [182]	213	China	SH and HR	16 to 20 weeks	SH = TSH > 97.50th percentile (4.21 mU/L), HR = tT4 < 2.50th percentile (101.79 nmol/L)	25–30 months	Bayley Scale of Infant Development
Henrichs et al., 2010 [183]	3659	The Netherlands	HR and Co TSH	13.3 weeks	HR = fT4 10th percentile (<11.76 pmol/L) and fifth percentile (<10.96 pmol/L), Co TSH = TSH ref range 0.03–2.50 mU/L	18 and 30 months	MacArthur Communicative Development Inventory at 18 months, Language Development Survey at 30 months
Suárez-Rodríguez et al., 2012 [94]	70	Spain	HR	37 weeks	fT4 < 10th percentile (9.5 pmol/L)	38 months and 5 years	The McCarthy Scales of Children’s Abilities
Williams et al., 2012 [184]	166	UK	SH and HR	±1 h after birth	SH = TSH > 3.0 mU/L, HR = fT4 ≤ 10th percentile (11.6 pmol/L) or tT4 ≤10th percentile (108.4 nmol/L)	5.5 years	The McCarthy Scales of Children’s Abilities
Craig et al., 2012 [185]	196	USA	HR	2nd trimester	fT4 < 3rd Percentile (11.84 pmol/L)	2 years	Bayley’s Scale of Infant Development III
Ghassabian et al., 2014 [93]/Korevaar et al., 2016 [96]	3737/5647	The Netherlands	HR and SH	13.5/13.2 weeks	HR = fT4 < 5th percentile (10.99 pmol/L), SH= TSH > 2.50 mU/L	6 years	Snijders-Oomen Niet-verbale intelligence test, revisie (Mosaics and Categories)
Päkkilä et al., 2015 [186]	5295	Finland	HR, SH and OH	Mean 10.7 weeks	HR = fT4 < 11.4–11.09 pmol/L depending upon trimester, SH = TSH > 3.10–3.50 mU/L depending upon trimester	8 and 16 years	Strengths and Weaknesses of ADHD Symptoms and Normal Behavior, Teacher reported child school performance, Youth Self Report and WISC-Revised
Grau et al., 2015 [187]	455	Spain	HR	1st and 2nd trimesters	<10th Percentile (13.7–11.5 pmol/L depending on trimester)	1 and 6–8 years	Brunet-Lezine scale and WISC-IV

Abbreviations: HR = Hypothyroxinemia, OH = Overt hypothyroidism, SH = Subclinical hypothyroidism, TFT = Thyroid function tests, Co = Continuous, TSH = Thyroid stimulating hormone.

## Data Availability

Not applicable.

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
