# Peer review of "Iodine Deficiency, Maternal Hypothyroxinemia and Endocrine Disrupters Affecting Fetal Brain Development: A Scoping Review"

_nutrients, 2023, doi:10.3390/nu15102249_

Round 1
Reviewer 1 Report
In all the articles that you have chosen according to your criteria, are all about the relationship between endocrine disruptors and thyroid hormones? I have seen that some only talk about the hormonal axis, or the iodine intake levels...I think you should check the PRISMA chart. Maybe some information could be a part of the Introduction instead of results.
Table 1. Please could you improve the abbreviation? For example, it is difficult to understand "Dates according to a[56], b[42], c[57], d[43], e[49], f[50, 51]."
Table 2. I think the aesthetics of the table could be improved.
Of the 279 papers very few appear in the tables, why?
Do endocrine disruptors interfere with the proper functioning of thyroid hormones even if iodine levels are sufficient? Perhaps the adequate iodine supply will not be enough, what other options would there be?
You have done a review in which you have read many articles (279!), although the conclusions you bring are well known and accepted. What is the novelty of your article compared to those already published?
Author Response
Response to Reviewer 1:
Thank you very much for your value review.
Your first point is related to the selection of the references:
Of course, not only articles directly dealing with a direct interaction of iodine uptake or organification, but also thyroid dysfunction or hormone action had been included. In many cases it is difficult, because the plenty interactions are not fully understood. But the specific effects of TDC are described in Tab 3, and is also discussed in detail in the accompanying discussion.
Concerning Tab1: the suffix a, b, c etc should help to find respective references more easily. Those data, with no suffix are drawn from Ref 41, and 42
Tab 2 should give a short overview on the current knowledge; the details are described in the text. We understand that the table is not easy to read, it is more to get an impression of the overwhelming already described effects of TDC.
The 279 papers are not only related to TDC, but also to iodine deficiency in Europe, during pregnancy and refer to the relation to hypothyroxinemia, as well as outcome of children born in areas where the mothers might have iodine deficiency.
Unfortunately, there still is no clear answer, whether more iodine intake might attenuate the TDC effects. Of course, the best would be to educed the distribution of TDC in our environment. Specific antidots, to our knowledge, are currently not available. More intensified scientific work is necessary.
Many things in this field have not been sufficiently clarified, though a lot of effort had been done, but until now without clear consequences. The effects iodine deficiency in the embryonal/fetal development have been known for a long time, and also the positive influence of USI, but as we demonstrated only few countries changed their legislative and have introduced USI. Similar holds true with respect to iodine supplementation in pregnant women, if they have low iodine intake. And the deleterious effects of TCD are known, but mainly among experts. The article also should focus on prevention again and again, because not enough had been done in Europe. It is also addressed to health authorities and think about precautious prevention.

Reviewer 2 Report
Very large scope review on iodine deficiency in early pregnancy and risk for neurologic development in European countries. In relationship with endocrine disrupters. Minor remarks:
1) Bottom page 3: "A mean UIC of 150–249 µg/L during pregnancy is assumed as an adequate iodine supply [39,40]": To verify: is it not the "median UIC" as indicator of IDD in pregnant women?
2) Table 3: please, give the translation of abbreviations in your legend: "HR", "SR", "TFT", "co-TSH", "OH"
3) Discussion 1: you should also mention the risk of excess iodine which justifies the very careful approach of "mandatory" iodine supplement in general population (increase of autoimmune thyroiditis, for example). Actually, your manuscript suggests that a targeted iodine supplement in pregnant women in European countries is the best strategy, and not a "mandatory" increase of iodine in whole population as the objectives of suffisient iodine intake in non pregnant women, in men and in children looks adequate (your table 1). So, if you agree with this point of view, you should be more clear in your discussion to limit this iodine supplement to pregnant women.
Discussion 2: suggestion: your manuscript puts also in evidence the requirement to develop a research program at the European dimension (and not more at the national level). It is worthwhile to insist on that point, by defining clearly objectives of such an international study involving European countries.
Author Response
Response to Reviewer 2
Thank you very much for reviewing our manuscript and the valuable queries.
- Thank you for this point. It is correct, the iodine intake of pregnant women should be between 150-250 µg per day and mean UIC 250 µg/L, we have corrected it.
- The abbreviations in Tab 3 are now explained
- Thank you for this point, which of course is important.
We include this in the conclusion: “The Krakow Declaration of Iodine, published by the EUthyroid consortium and other organizations raised major points how iodine deficiency can be efficiently eradicated in Europe. It was demanded that (1) universal salt iodization should be harmonized across European countries, (2) regular monitoring and evaluation studies have to be established to continuously measure the benefits and potential harms of iodine fortification programs, and (3) societal engagement is needed to warrant sustainability of IDD prevention programs.
Benefits and Harms of a Prevention Program for Iodine Deficiency Disorders: Predictions of the Decision-Analytic EUthyroid Model.
Schaffner M, Mühlberger N, Conrads-Frank A, Qerimi Rushaj V, Sroczynski G, Koukkou E, Heinsbaek Thuesen B, Völzke H, Oberaigner W, Siebert U, Rochau U.Thyroid. 2021 Mar;31(3):494-508. doi: